# Octopamine and tyramine signalling in *Aedes aegypti*: Molecular characterization and insight into potential physiological roles

**Luca Finetti**[1]*, **Jean-Paul Paluzzi**[2], **Ian Orchard**[1], **Angela B. Lange**[1]

**1** Department of Biology, University of Toronto Mississauga, Mississauga, ON, Canada, **2** Department of Biology, York University, Toronto, ON, Canada

* luca.finetti@utoronto.ca

**Data Availability Statement:** All relevant data are within the paper and its Supporting Information files.

## Abstract

In insects, the biogenic amines octopamine (OA) and tyramine (TA) are involved in controlling several physiological and behavioural processes. OA and TA act as neurotransmitters, neuromodulators or neurohormones, performing their functions by binding to specific receptors belonging to the G protein-coupled receptor (GPCR) superfamily. OA and TA along with their receptors are involved in reproduction, smell perception, metabolism, and homeostasis. Moreover, OA and TA receptors are targets for insecticides and antiparasitic agents, such as the formamidine Amitraz. In the dengue and yellow fever vector, *Aedes aegypti*, limited research has been reported on their OA or TA receptors. Here, we identify and molecularly characterize the OA and TA receptors in *A. aegypti*. Bioinformatic tools were used to identify four OA and three TA receptors in the genome of *A. aegypti*. The seven receptors are expressed in all developmental stages of *A. aegypti*; however, their highest transcript abundance is observed in the adult. Among several adult *A. aegypti* tissues examined, including the central nervous system, antennae and rostrum, midgut, Malpighian tubules, ovaries, and testes, the type 2 TA receptor (TAR2) transcript is most abundant in the ovaries and the type 3 TA receptor (TAR3) is enriched in the Malpighian tubules, leading us to propose putative roles for these receptors in reproduction and diuresis, respectively. Furthermore, a blood meal influenced OA and TA receptor transcript expression patterns in adult female tissues at several time points post blood meal, suggesting these receptors may play key physiological roles associated with feeding. To better understand OA and TA signalling in *A. aegypti*, the transcript expression profiles of key enzymes in their biosynthetic pathway, namely tyrosine decarboxylase (*Tdc*) and tyramine β-hydroxylase (*Tβh*), were examined in developmental stages, adult tissues, and brains from blood-fed females. These findings provide information for better understanding the physiological roles of OA, TA, and their receptors in *A. aegypti*, and additionally, may help in the development of novel strategies for the control of these human disease vectors.

**Funding:** This research was supported by NSERC Discovery Grants to JPP (RGPIN-2020-061130), IO (RGPIN-2017-06402) and ABL (RGPIN-2019-05775). The funders had no role in study design, data collection and analysis, decision to publish, or preparation of the manuscript.

**Competing interests:** The authors have declared that no competing interests exist.

## Introduction

*Aedes aegypti* is the primary vector for dengue, chikungunya, and yellow fever viruses, which cause debilitating diseases that together are responsible for hundreds of millions of infections and thousands of deaths, annually, worldwide [1, 2]. Although localized mainly in tropical areas, these diseases are more broadly distributed due to increasingly frequent international travel [3, 4]. Furthermore, climate change is responsible for creating new territories where *A. aegypti* can become endemic [5]. Mosquito-borne viral infections can be prevented by boosting host immunity (vaccines) or controlling the vector [6]. The principal intervention strategies include the use of insecticides (such as pyrethroids), genetic techniques (sterile male) or using Wolbachia infected mosquitoes [7]. However, all of these tools, along with vaccination campaigns, have not resolved the burden these pests impose on humans, and therefore, it is necessary to identify new targets for the control of *A. aegypti*. For this reason, understanding the *A. aegypti* physiology is critical for identifying new molecular targets for design of lead compounds to eradicate these pests. In insects, the biogenic amines dopamine, serotonin, tyramine and octopamine play important roles in controlling various physiological processes [8–10]. In *A. aegypti*, both serotonin and dopamine are implicated in blood feeding behavior and development [11–13]; however, the roles of OA and TA in *A. aegypti* physiology have not been studied.

In insects, TA and OA control and modulate a broad range of biological functions essential for life [14]. To synthesize OA and TA, tyrosine is decarboxylated by tyrosine decarboxylase (Tdc), which gives rise to TA. This can then be hydroxylated to OA by tyramine β-hydroxylase (Tβh) [15]. The insect nervous system contains high levels of both OA and TA, suggesting that they can each act as neurotransmitters. Moreover, they also appear to act as neuromodulators and neurohormones in a wide variety of physiological processes, and can operate in a paracrine, endocrine, and autocrine fashion in peripheral organs [16, 17]. Originally, TA was considered only as the intermediate product necessary for the synthesis of OA [18]. However, it is now well established that TA and OA perform important functions, as neurotransmitters, neurohormones and neuromodulators, independently of each other [19]. In many cases, TA and OA operate as antagonistic modulators in a coordinated way to control processes such as olfaction, locomotion, metabolism, fertilization, and reproduction [20–23].

TA and OA exert their physiological actions by interacting with and activating different receptors, the tyramine (TAR) and the octopamine (OAR) receptors [24]. The OA and TA receptors are classified into different groups based on their primary structure and the intracellular pathway activated [25, 26], resulting in two OA α-adrenergic-like receptors (OAα1-R and OAα2-R), three OA β-adrenergic-like receptors (OAβ1-R, Oaβ2-R and OAβ3-R) and three TA receptors (TAR1, TAR2 and TAR3). The study of TA and/or OA receptor-deficient insects has revealed that the corresponding receptors play important roles in modulating their biology, physiology, and behavior [27]. Thus, changing the normal functioning of these receptor classes by either blocking or overstimulating them can lead to death of the insect or interference with their fitness and reproductive capacity [28]. It is therefore not surprising that both OA and TA receptors have proven to be interesting targets for insecticides. Amitraz is an acaricide and non-systemic insecticide that targets OA receptors [29]. However, recent studies have demonstrated that amitraz can also exert its toxic effect through TAR1 [30, 31]. In addition, TA and OA signalling pathways are interesting targets for natural insecticides such as monoterpenes [32–34].

Currently, few studies have been published on mosquito TA and OA receptors providing limited insight on their roles and expression. The effect of TA, OA and several biogenic amine receptor inhibitors on egg production was investigated in *A. gambiae* [35] and two OA receptors were functionally characterized [36].

In this work, we have identified and characterized OA and TA receptors from the genome of *A. aegypti* and have investigated their transcript expression profiles over all development stages, and several tissues from male and female adults. Furthermore, we set out to determine if a blood meal influences OA and TA receptor expression patterns in various tissues of the adult female, which could provide insight on OA and TA signalling critical for physiological processes following blood meal engorgement.

## Materials and methods

### *A. aegypti*: Rearing and feeding

Eggs of *A. aegypti* (Liverpool strain) were collected from an established colony maintained at York University, Canada [37]. All mosquitoes were raised under a 12:12 light: dark cycle at 26˚C as previously described [38]. All adult mosquitoes were fed with 10% sucrose (w/v) *ad libitum* and used for experiments three days-post ecdysis. For blood-feeding studies, three-day old post-ecdysis females were fed with sterile rabbit's blood (Cedarlane Laboratories, Burlington, ON, Canada) following Sri-in and colleagues [39].

### Bioinformatics

OA and TA receptors from *Drosophila melanogaster* (Accession numbers: DmOAα1-R, NM_169914.2; DmOAα2-R, NM_0012755785.2; DmOAβ2-R, NM_001316576.1; DmOAβ3-R, NM_001038954.2; DmTAR1, NM_079695.4; DmTAR2, NM_001300453.1; DmTAR3, NM_001300451.1) were used as query in BLASTN searches on VectorBase with the most recent *A. aegypti* genome database (Accession: AaegL5.0) to identify orthologous transcripts that code for putative OA and TA receptors.

Total RNA was extracted from five adult mosquitoes using TRIzol™ Reagent (Thermo Fisher Scientific, USA), quantified in a micro-volume spectrophotometer and analysed by 0.8% w/v agarose gel electrophoresis. Total RNA (1 μg) was treated with DNase I (Thermo Fisher Scientific, USA) and used for the synthesis of cDNA, carried out with a High-Capacity cDNA Reverse Transcription Kit (Thermo Fisher Scientific, USA). For amplification of the full *A. aegypti* putative OA and TA receptor open reading frames (ORFs), specific primers were designed based on the annotated sequences (S1 Table). High fidelity amplification was achieved using Herculase II Fusion DNA Polymerase (Agilent, USA) and a touchdown thermal cycling program: initial denaturation at 95˚C for 3 min, followed by 10 cycles at 95˚C for 20 s, 66–56˚C for 30 s (minus 1˚C/cycle), 68˚C for 4 min, 25 cycles at 95˚C for 20 s, 56˚C for 30 s, 68˚C for 4 min and a final extension at 68˚C for 5 min. PCR products were directly sequenced at Eurofins Genomics (Toronto, ON, Canada). The sequence obtained were used for subsequent *in silico* and molecular analysis.

### Multiple sequence alignment and bioinformatic analysis

Phylogenetic maximum likelihood analysis was performed using MEGA software (version 11) with 500-fold bootstrap resampling. The *D. melanogaster* odorant receptor Orco (OR83b) was used as an outgroup to root the tree (NP_001097687.1). Multiple protein sequence alignments between the deduced amino acid sequence of *A. aegypti* OA and TA receptors and other biogenic amine receptor sequences were performed using Clustal Omega (https://www.ebi.ac.uk/Tools/msa/clustalo/) and BioEdit Sequence Alignment Editor 7.2.6.1. Molecular characterization of the putative *A. aegypti* OA and TA receptors were carried out using TMHMM v.2.0 Server and the Kyte and Doolittle method [40]. Potential N-glycosylation sites were predicted by NetNGlyc 1.0 Server (https://services.healthtech.dtu.dk/service.php?NetNGlyc-1.0) while

potential targets for protein kinase A and protein kinase C phosphorylation were identified using the NetPhos 3.1 Server (https://services.healthtech.dtu.dk/service.php?NetPhos-3.1).

### Quantitative real-time PCR analysis

Total RNA was extracted from whole bodies of *A. aegypti* at various developmental stages (for each replicate: 100 mg of eggs, twenty 1st instar larvae, ten 2nd or 3rd instar larvae, five 4th instar larvae, three pupae, and three adults) and different tissues/organs (at least twenty animals were dissected for each biological replicate) using TRIzol™ Reagent (Thermo Fisher Scientific). The tissues of *A. aegypti* were dissected under sterile cold phosphate buffered saline (PBS, 6.6 mM Na2HPO4/KH2PO4, 150 mM NaCl, pH 7.4). Five hundred ng of purified total RNA was then treated with DNase I (Thermo Fisher Scientific) and used for cDNA synthesis, performed using a High-Capacity cDNA Reverse Transcription Kit (Thermo Fisher Scientific, USA). The qPCR analyses were performed using a 384 CFX Connect Real-Time PCR Detection System (Bio-Rad, USA) in a 10 μL reaction mixture containing 4.2 μL of 1:5 diluted cDNA, 5 μL of Advanced qPCR master mix (Wisent Bioproducts Inc., QC, Canada), 0.4 μL forward primer (10 μmol l$^{-1}$) and 0.4 μL reverse primer (10 μmol l$^{-1}$). Thermal cycling conditions were 95˚C for 2 min, 40 cycles at 95˚C for 15 s and 60˚C for 30 s. After the cycling protocol, a melt-curve analysis from 60 to 95˚C was applied. Expression of *A. aegypti* OA/TA receptors was normalized in accordance with the relative quantification method [41] or $2^{-\Delta\Delta Ct}$ method [42] using *Actin*, *Rps17* and *Rpl32* as reference genes [43]. Gene-specific primers (S1 Table) were used and at least three independent biological replicates, tested in duplicate, were performed for each sample.

### Statistics

All data were examined using Graph Pad Prism 6.0 (La Jolla, CA, USA). Data are expressed as means ± s.e.m. of n experiments and were analyzed using one-way analysis of variance (ANOVA) followed by Bonferroni test for multiple comparison.

## Results

### Molecular characterization of *A. aegypti* OA and TA receptors

*A. aegypti* OA and TA receptors were identified by multiple sequence homology using orthologous OA and TA receptors from *D. melanogaster*. A total of four putative OA receptors and three putative TA receptors were identified in both NCBI and VectorBase datasets in the *A. aegypti* genome: XP_021695040.1, AGX85004.1, XP_021693342.1, XP_021693338.1, XP_001652255.3, XP_021692997.1 and XP_021692998.1. The amino acid sequence of several insect biogenic amine receptors were used to construct a phylogenetic tree by maximum likelihood analysis and the results indicated that each putative *A. aegypti* OA/TA receptor clusters in specific clades (Fig 1).

This phylogenetic analysis suggests XP_021695040.1, AGX85004.1, XP_021693342.1 and XP_021693338.1 encode for OA receptors, while XP_001652255.3, XP_021692997.1 and XP_021692998.1 encode for TA receptors. Among the OA receptors, we identified two α-adrenergic-like OA receptors (XP_021695040.1 and AGX85004.1) and two β-adrenergic-like OA receptors (XP_021693342.1 and XP_021693338.1). Interestingly, the sequence AGX85004.1 is annotated as a putative neuropeptide Y-like receptor (NPYLR6) sequence characterized in *A. aegypti* by Liesch and colleagues [44] although several neuropeptides that were tested failed to activate this receptor. The current phylogenetic analysis suggests that the *A. aegypti* NPYLR6 is a member of the insect OAα2 receptor family.

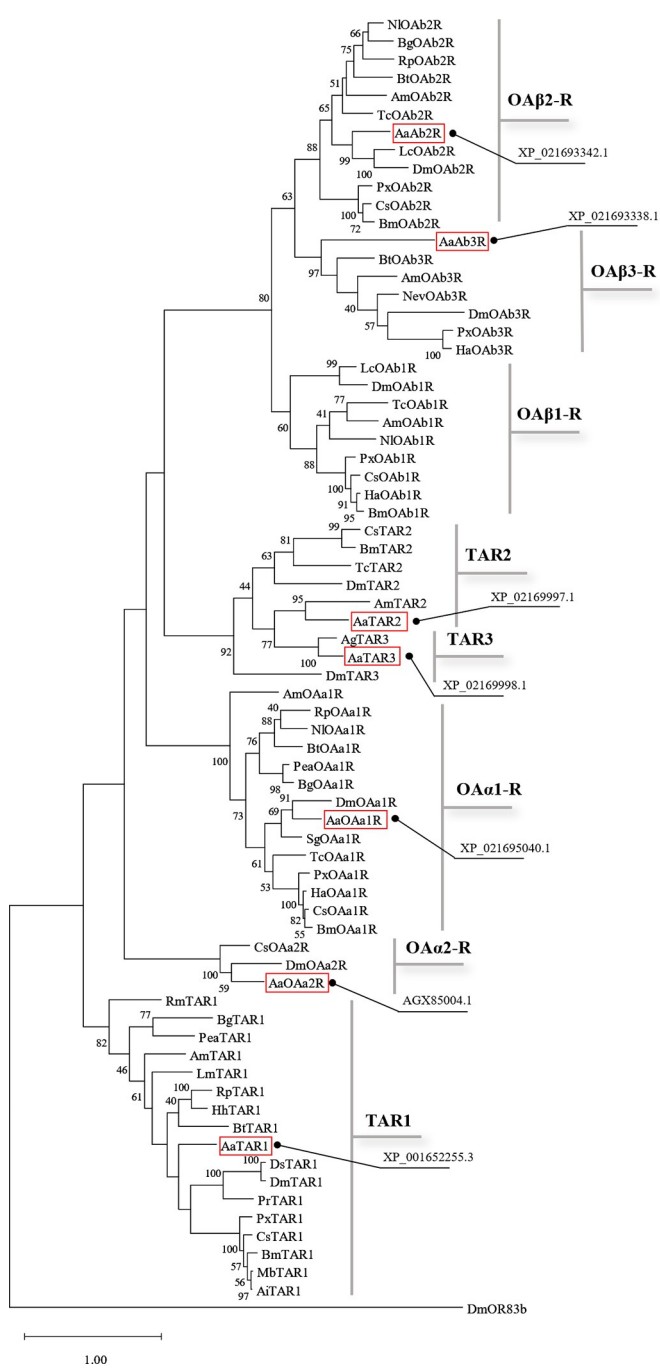

**Fig 1. Phylogenetic relationship determined using maximum likelihood analysis of *A. aegypti* OA and TA receptors and other insect amine receptors.** The characterized *A. aegypti* OA/TA receptors are outlined in red. The values shown at the nodes of the branches are the percentage bootstrap support (500 replications) for each branch. Alignment was performed using the amino acid sequences found in GenBank (accession number for each sequence is noted and reported in S2 Table). *D. melanogaster* OR83b receptor was chosen as outgroup.

## *A. aegypti* OA receptors

The analysis of the amino acid sequences of the four putative *A. aegypti* OA receptors revealed that the largest protein was encoded by *AaOAβ3* (1196 aa) followed by *AaOAα2* (617 aa),

**Table 1. OA receptors identified and characterized in *A. aegypti*.**

| Nucleotide accession | XM_021839348.1 | KC439536.1 | XM_021837650.1 | XM_021837643.1 |
|---|---|---|---|---|
| Protein accession | XP_021695040.1 | AGX85004.1 | XP_021693342.1 | XP_021693338.1 |
| Receptor name | **AaOAα1-R** | **AaOAα2-R** | **AaOAβ2-R** | **AaOAβ3-R** |
| Coding DNA sequence (CDS) (bp) | 4932 | / | 3315 | 4201 |
| Open reading frame (ORF) (bp) | 1797 | 1854 | 1638 | 3588 |
| Protein length (aa) | 598 | 617 | 545 | 1193 |
| Predicted molecular weight (KDa) | 65.5 | 66.55 | 59.41 | 130.1 |
| Protein kinase C (PKC) putative phosphorylation sites | $S^{48}$, $S^{179}$, $T^{188}$, $T^{211}$, $T^{217}$, $S^{221}$, $T^{284}$, $T^{294}$, $S^{317}$, $T^{352}$, $T^{354}$, $T^{469}$ | $S^{28}$, $T^{94}$, $T^{279}$, $S^{324}$, $T^{391}$ | $S^{26}$, $T^{53}$, $S^{74}$, $T^{79}$, $S^{211}$, $T^{259}$, $T^{288}$, $T^{289}$, $T^{330}$, $T^{404}$, $T^{428}$ | $S^{118}$, $T^{243}$, $T^{396}$, $S^{439}$, $S^{458}$, $S^{464}$, $S^{483}$, $S^{509}$, $T^{523}$, $T^{538}$, $S^{556}$, $S^{561}$, $S^{593}$, $T^{687}$, $T^{782}$, $T^{793}$, $T^{888}$, $S^{936}$, $S^{939}$, $S^{940}$ |
| Protein kinase A (PKA) putative phosphorylation sites | $S^{380}$ | $S^{433}$ | $S^{151}$, $T^{488}$, $T^{501}$, $S^{511}$, $S^{512}$, $S^{521}$ | $S^{298}$, $S^{326}$, $S^{381}$, $S^{384}$, $S^{401}$, $S^{419}$, $S^{435}$, $S^{772}$, $S^{983}$, $S^{1076}$ |
| N-glycosylation putative sites | $N^2$, $N^{174}$ | $N^{15}$, $N^{142}$ | $N^{48}$, $N^{58}$ | $N^{39}$, $N^{53}$, $N^{74}$ |

*AaOAα1* (598 aa) and *AaOAβ2* (545 aa) (Table 1). The long amino acid sequence of the *AaOAβ3* receptor was due (in part) to a long third intracellular loop (823 residues) between transmembrane (TM) V and TM VI, a feature conserved also in the *D. melanogaster* OAβ3 receptor (841 residues) [45]. Characteristic features of GPCRs were identified in all *A. aegypti* OA receptors (S1–S4 Figs), such as the seven transmembrane domains and a DRY domain located in the TM III, required for G-protein/receptor binding [46]. Furthermore, for each receptor, potential N-glycosylation and phosphorylation sites, possibly necessary for receptor regulation, were identified (Table 1). Multiple sequence alignment was performed for each *A. aegypti* OA receptor and a high degree of conservation was observed mainly in the regions corresponding to the predicted seven transmembrane domains of the protein structure (S1–S4 Figs). Furthermore, an aspartic acid residue in TM III and several serine residues in TM V were identified for each *A. aegypti* OA receptor; residues mainly implicated in biogenic amine binding (S1–S4 Figs).

## *A. aegypti* TA receptors

The three identified *A. aegypti* TA receptors were 520 (AaTAR1), 720 (AaTAR2) and 543 (AaTAR3) amino acid residues long (Table 2). All three TA receptors contained seven transmembrane domains and the DRY domain in TM III (S5–S7 Figs). Moreover, all the three *A. aegypti* TA receptor sequences contained potential protein kinase A and protein kinase C phosphorylation sites as well as N-glycosylation residues (Table 2). From the multiple sequence alignments, a high degree of conservation occurred in regions corresponding to the predicted seven transmembrane domains of each TA receptor (S5–S7 Figs). With regards to the interaction between receptor and biogenic amines, well conserved aspartic acid residues were identified in TM III as well as several serine residues in TM V for each *A. aegypti* TA receptor (S5–S7 Figs).

## Transcript expression pattern in developmental stages

To better understand the involvement of OA and TA receptors in *A. aegypti* growth and development, their expression profiles were investigated in all *A. aegypti* developmental stages (egg, 1st to 4th larvae, pupae, and adults). The analysis revealed that the seven receptors were expressed in all development stages (Fig 2). In eggs, *AaOAβ3-R* showed the highest transcript

**Table 2. TA receptors identified and characterized in *A. aegypti*.**

| Nucleotide accession: | XM_001652205.3 | XM_021837305.1 | XM_021837306.1 |
|---|---|---|---|
| Protein accession: | XP_001652255.3 | XP_021692997.1 | XP_021692998.1 |
| Receptor name: | AaTAR1 | AaTAR2 | AaTAR3 |
| Coding DNA sequence (CDS) (bp) | 3823 | 3657 | 1848 |
| Open reading frame (ORF) (bp) | 1563 | 2163 | 1632 |
| Protein length (aa) | 520 | 720 | 543 |
| Protein molecular weight (KDa) | 57.46 | 77.34 | 60.39 |
| Protein kinase C (PKC) putative phosphorylation sites | $T^{254}$, $S^{265}$, $T^{269}$, $S^{278}$, $S^{315}$, $S^{361}$, $S^{372}$, $T^{395}$, $T^{408}$, $T^{413}$, $S^{420}$, $T^{448}$ | $S^{233}$, $S^{234}$, $S^{239}$, $T^{315}$, $T^{320}$, $T^{400}$, $S^{408}$, $S^{431}$, $S^{437}$, $S^{439}$, $T^{446}$, $T^{483}$, $S^{499}$, $T^{503}$, $T^{505}$, $S^{506}$, $S^{705}$, $S^{717}$ | $T^{224}$, $S^{226}$, $T^{303}$, $T^{332}$, $T^{338}$, $S^{346}$, $S^{360}$, $T^{374}$, $T^{454}$ |
| Protein kinase A (PKA) putative phosphorylation sites | $S^{265}$, $S^{438}$ | $T^{239}$, $S^{438}$, $S^{439}$, $S^{543}$, $S^{551}$, $T^{583}$ | $T^{150}$, $S^{448}$ |
| N-glycosylation putative sites | $N^{24}$ | $N^{313}$ | $N^{24}$, $N^{53}$ |

abundance (Fig 2A), whereas *AaOAα1-R* had the highest transcript levels throughout the larval stages, suggesting a more prominent role in larval biology compared to the other OA and TA receptors (Fig 2B–2E). In the pupal stage, *AaTAR3* transcript levels trended higher compared to all the other receptors, but the increase was not statistically different (Fig 2F). Male and female adults showed a different expression pattern from each other. In adult females, *AaTAR2* had the highest transcript abundance while in males no differences were observed among all the receptors (Fig 2G and 2H). However, as an overall observation of enrichment of these multiple receptors, *A. aegypti* adult males exhibited the highest receptor transcript expression levels compared to the other development stages (Fig 2I). Interestingly, except for *AaOAα1-R* that had similar transcript levels across larval and adult stages, all the other receptors showed a reduction in transcript abundance during the larval stages (Fig 2I).

## Transcript expression pattern in adult tissues

To better understand the role of OA and TA receptors in controlling physiology and fitness of *A. aegypti* adults, their transcript abundance levels were quantified in different adult tissues of both sexes: brain, antennae and rostrum combined, Malpighian tubules (MTs), midgut and the reproductive organs, ovaries and testes. In female brains, the TA receptors *AaTAR1* and *AaTAR3* showed significantly enriched mRNA abundance in comparison to the other receptors (Fig 3A). In male brains, *AaTAR3* also demonstrated significant enrichment compared to other receptor transcripts (Fig 3B). In antennae and rostrum, females did not show any significant differences in the transcript levels among all the receptors (Fig 3C). Conversely, in males *AaOAβ2-R* was the receptor most highly expressed, suggesting a role in processes associated with smell, taste and hearing (Fig 3D). In both the MTs and the midgut of both sexes, *AaTAR3* was the receptor with the highest transcript abundance compared to the other receptors (Fig 3E–3H). With regards to the reproductive organs, *AaTAR2* was significantly enriched in the ovaries (Fig 3I) whereas no significant differences in transcript levels among all the receptors was observed in the testes (Fig 3L).

## TA and OA receptor transcript expression patterns after a blood meal

Female *A. aegypti* require a blood meal for egg production and it is also true that this process imposes some stress on the insect. Indeed, the ingestion of blood requires the activation of several physiological events associated with blood digestion and nutrient absorption, but also elimination of toxins and excess water and salts. To better understand the roles of OA and TA receptors in these and other complex events, the temporal expression of all seven receptors

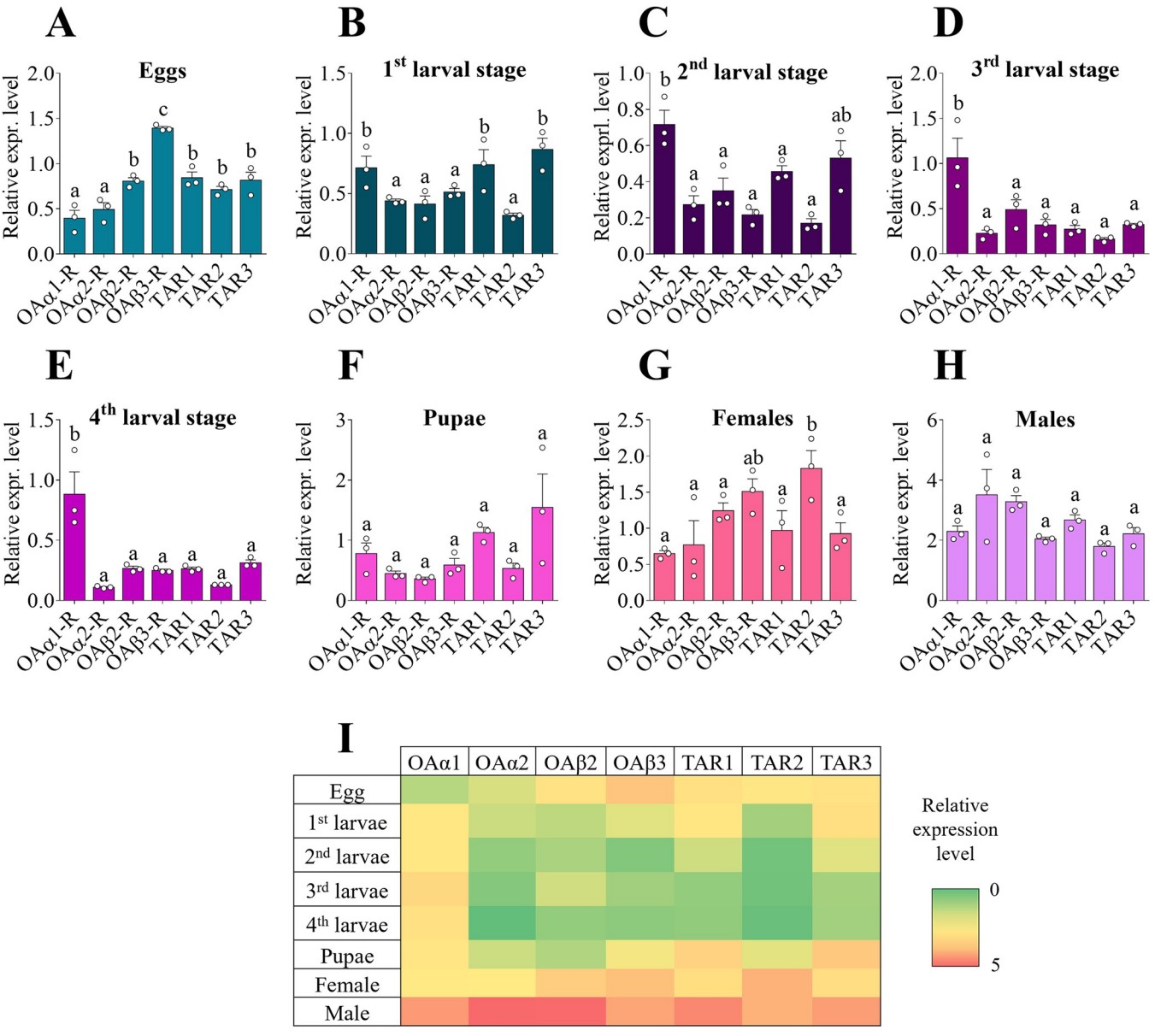

**Fig 2. Expression patterns of *A. aegypti* OA and TA receptor transcripts over development stages.** Eggs (**A**), 1st larval stage (**B**), 2nd larval stage (**C**), 3rd larval stage (**D**), 4th larval stage (**E**), pupae (**F**), adult female (**G**) and adult male (**H**). The data obtained from all the development stages is also represented as a heat map (**I**). Data represent means ± SEM of three biological replicates. One-way ANOVA followed by multiple comparisons Bonferroni post hoc test were performed. Statistical significance of p<0.05 is indicated by different letters in panels A-H.

was evaluated along five time points after blood feeding: 1, 6, 12, 24 and 48 hours post blood meal (PBM). Moreover, receptor transcript expression was investigated in four different tissues: brain, midgut, MT, and ovaries.

**Brain.** No significant difference between pre- and post-blood meal was observed for all the receptors investigated in the brain. *AaOAα1-R* showed a slightly increased expression at 12, 24 and 48 hours PBM (Fig 4A and 4H) whereas *AaOAα2-R* transcript underwent a slight reduction between 24 and 48 hours PBM (Fig 4B and 4H). Both *AaOAβ2-R* and *AaOAβ3-R* were similarly expressed across all examined time points (Fig 4C, 4D and 4H). In comparison

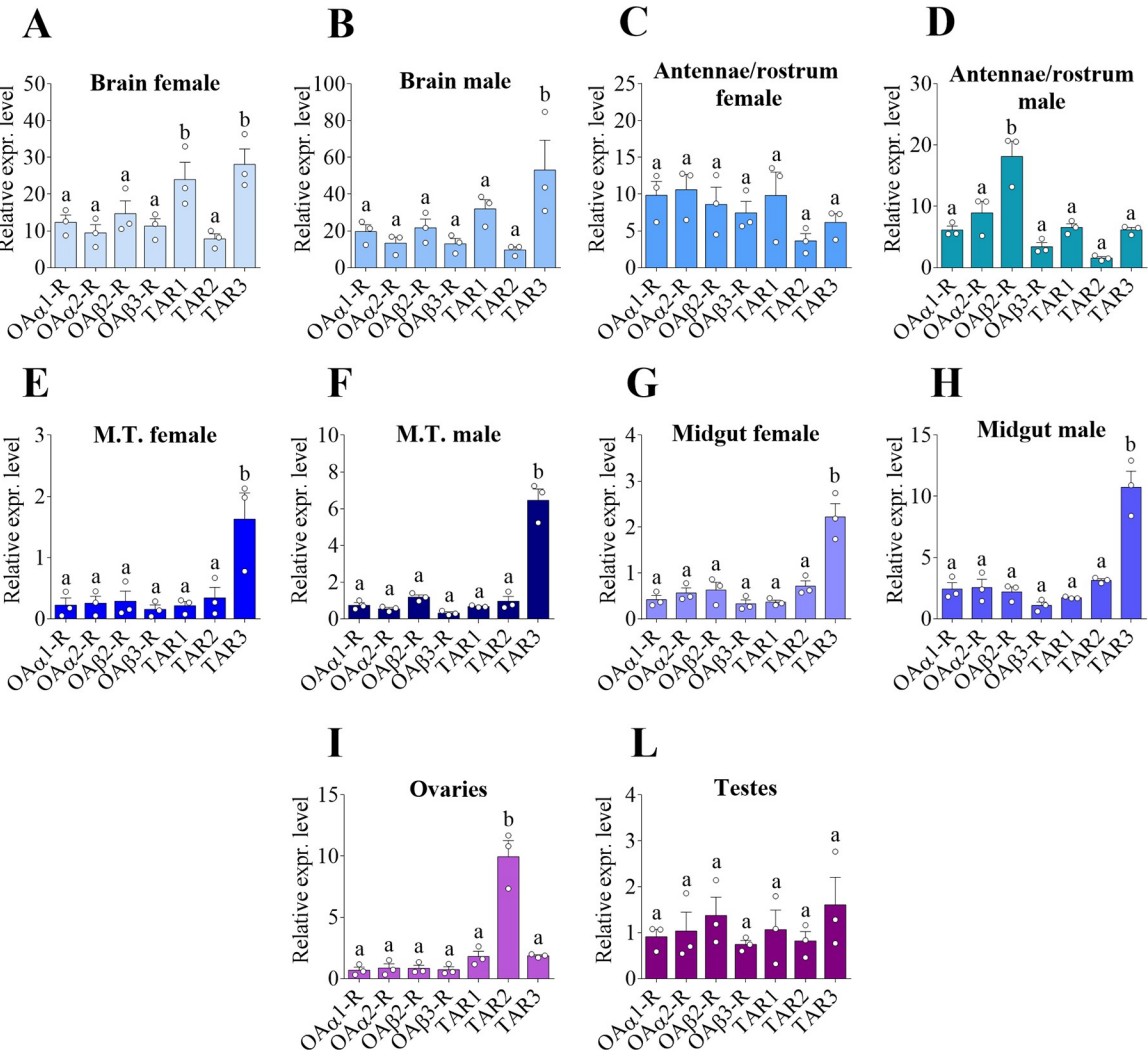

**Fig 3. Expression pattern of OA and TA receptor transcripts among female and male adult *A. aegypti* tissues.** Brain (**A, B**), antennae and rostrum (**C, D**), Malpighian tubules (MTs) (**E, F**), midgut (**G, H**) ovaries (**I**) and testes (**L**). Data represent means ± SEM of three biological replicates. One-way ANOVA followed by multiple comparisons Bonferroni post hoc test were performed. Statistical significance of $p < 0.05$ is indicated by different letters in panels A-L.

to pre blood-fed conditions, all the TA receptors exhibited a general slight downregulation starting from 12 hours PBM, with the most prominent mRNA reduction occurring between 24 and 48 hours PBM (Fig 4E–4H).

**Midgut.** In mosquitoes, and in general in all blood-feeding insects, the midgut is the portion of the digestive tract responsible for blood digestion, water regulation, digestive enzyme synthesis and secretion, nutrient absorption, and immunity [47]. *AaOAα1-R* showed an expression level trending towards an increase from 12 hours PBM, with the highest abundance at 48 hours PBM (Fig 5A and 5H). *AaOAα2-R* abundance trended towards a reduced level starting from 1 hour PBM but then exhibited higher expression levels 48 hours PBM (Fig 5B and 5H). *AaOAβ2-R* abundance did not change at 1, 6 and 48 hours PBM, but levels trended lower at 12 and 48 hours PBM (Fig 5C and 5H). The expression profile of *AaOAβ3-R* demonstrated levels trending lower between 1–24 hours PBM with lowest transcript abundance at 6 hours PBM (Fig 5D and 5H). The transcript expression profile for *AaTAR1* was similar to

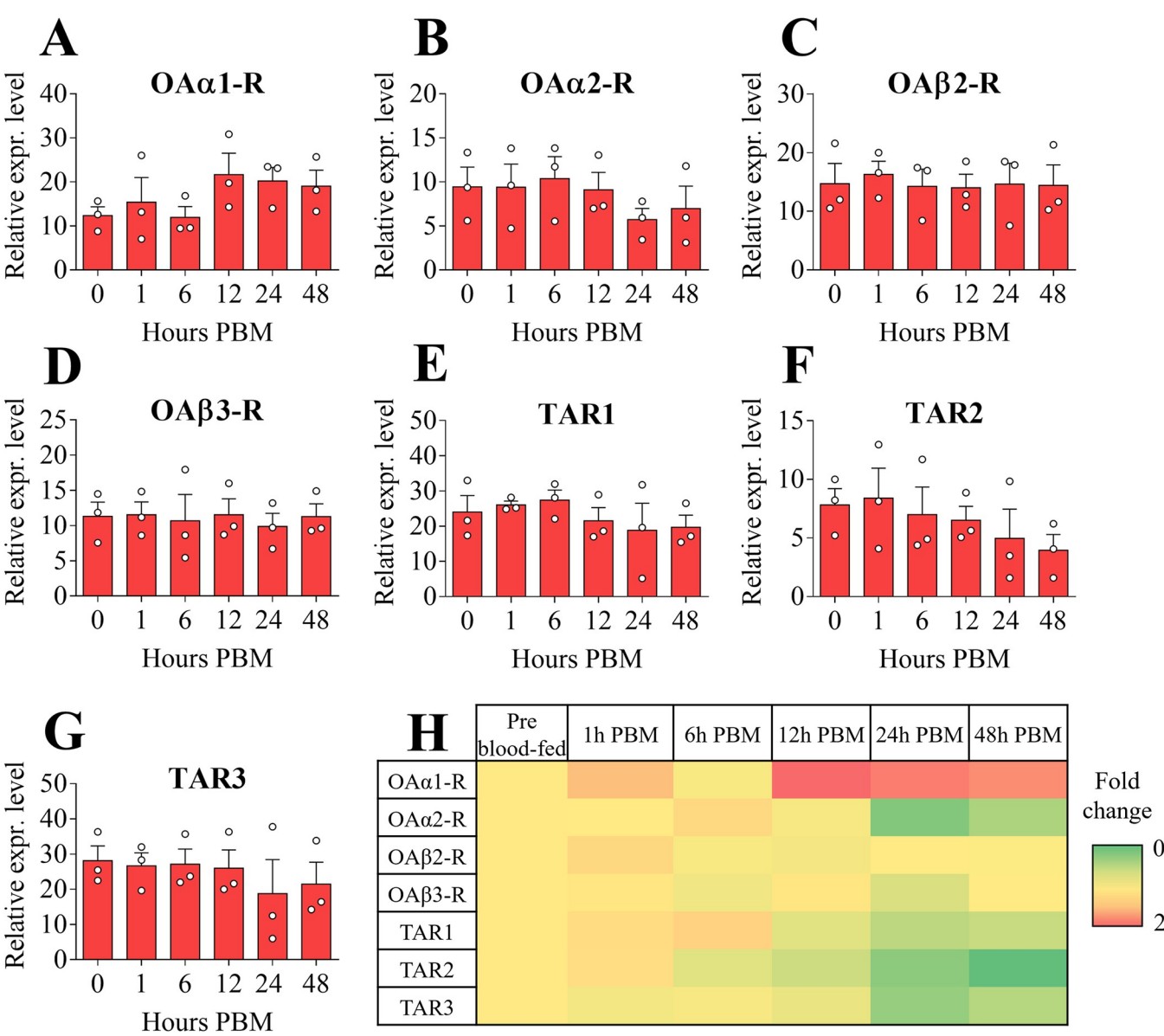

**Fig 4. Expression patterns of OA and TA receptor transcripts among female adult *A. aegypti* brain post blood meal (PBM) time points.** *AaOAα1-R* (**A**), *AaOAα2-R* (**B**), *AaOAβ2-R* (**C**), *AaOAβ3-R* (**D**), *AaTAR1* (**E**), *AaTAR2* (**F**) and *AaTAR3* (**G**). Normalizing receptor expression levels in pre blood-fed condition to 1, a heat map was also assembled (**H**). Data represent means ± SEM of three biological replicates. Statistical analysis was performed according to one-way ANOVA followed by multiple comparisons Bonferroni post hoc test.

*AaOAβ3-R* with lower abundance observed between 1–24 hours PBM (Fig 5E and 5H). Compared to the pre blood-fed condition, *AaTAR2* showed significantly increased transcript levels at 6 hours PBM, followed by a reduction from 12 hours PBM (Fig 5F and 5H). Interestingly, transcript abundance of *AaTAR3* was increased after blood feeding overall, with a significant increase in transcript abundance at 12 hours PBM, relative to the pre blood-fed condition (Fig 5G and 5H).

**Malpighian tubules.** In insects, MTs are the organs mainly implicated in excretion and osmoregulation. Since the blood meal is rich in elements that must be excreted, namely water and salts, MTs are one of major organs involved in this process [48]. *AaOAα1-R* exhibited a

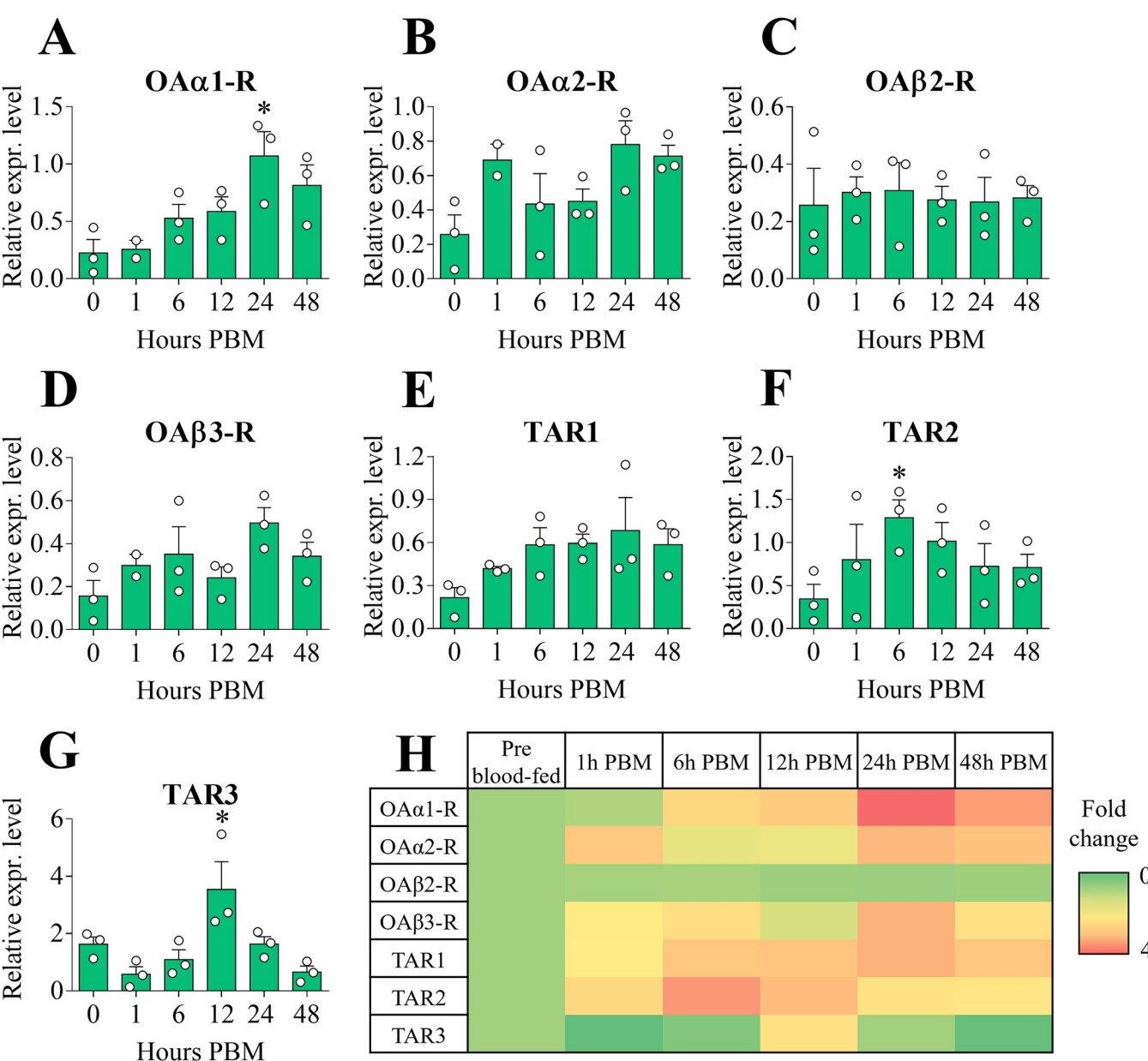

**Fig 5. Expression patterns of OA and TA receptor transcripts among female adult *A. aegypti* midgut post blood meal (PBM) time points.** *AaOAα1-R* (**A**), *AaOAα2-R* (**B**), *AaOAβ2-R* (**C**), *AaOAβ3-R* (**D**), *AaTAR1* (**E**), *AaTAR2* (**F**) and *AaTAR3* (**G**). Setting all the receptor expression levels in pre blood-fed condition to 1, a heat map was assembled (**H**). Data represent means ± SEM of three biological replicates. * $p<0.05$ vs pre blood-fed condition (0 hours PBM) according to one-way ANOVA followed by multiple comparisons Bonferroni post hoc test.

significant increase in transcript levels 24 hours PBM compared to the pre blood-fed condition and a trend in increased levels were also observed at 6, 12 and 48 hours PBM (Fig 6A and 6H). Although not significant, *AaOAα2-R* mRNA abundance trended higher at 1, 24 and 48 hours PBM (Fig 6B and 6H). *AaOAβ2-R* was stably expressed across all examined time points (Fig 6C and 6H) while *AaOAβ3-R* had the highest abundance at 24 hours PBM although this was not significantly different from the pre blood-fed condition (Fig 6D and 6H). *AaTAR1* abundance trended higher over all the time points but was not significantly different from transcript levels in non-blood fed females (Fig 6E and 6H). Compared to pre blood-fed

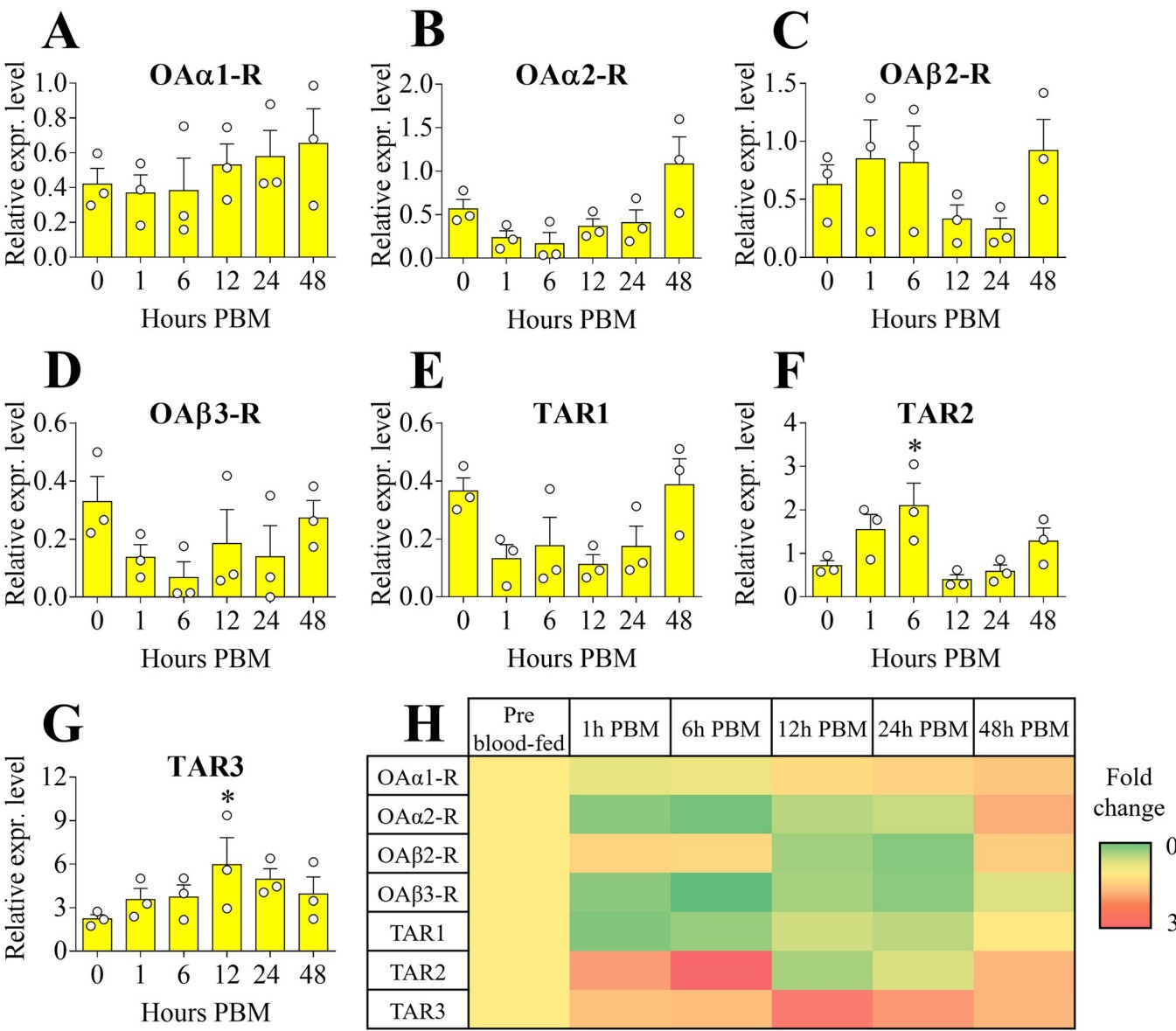

**Fig 6. Expression patterns of OA and TA receptor transcripts among female adult *A. aegypti* Malpighian tubules post blood meal (PBM) time points.** *AaOAα1-R* (**A**), *AaOAα2-R* (**B**), *AaOAβ2-R* (**C**), *AaOAβ3-R* (**D**), *AaTAR1* (**E**), *AaTAR2* (**F**) and *AaTAR3* (**G**). Setting all the receptor expression levels in pre blood-fed condition to 1, a heat map was assembled (**H**). Data represent means ± SEM of three biological replicates. * p<0.05 vs pre blood-fed conditions (0 hours PBM) according to one-way ANOVA followed by multiple comparisons Bonferroni post hoc test.

condition, *AaTAR2* showed increased transcript levels across all the time points analyzed, with a significant enrichment at 6 hours PBM (Fig 6F and 6H). On the other hand, *AaTAR3* exhibited a significant transcript accumulation in the MTs at 12 hours PBM (Fig 6G and 6H).

**Ovaries.** In the ovaries, the organ involved in egg production, all OA receptors showed a general reduction in expression levels across the examined time points, in comparison to pre blood-fed animals. Although not significant, *AaOAα1-R* abundance demonstrated the most prominent reduction at 48 hours PBM (Fig 7A and 7H). Both *AaOAα2-R* and *AaOAβ2-R*, showed a significant reduction in their transcript quantity 24 hours PBM with a similar trend in reduction for each receptor at 48 hours PBM (Fig 7B, 7C and 7H) while *AaOAβ3-R*

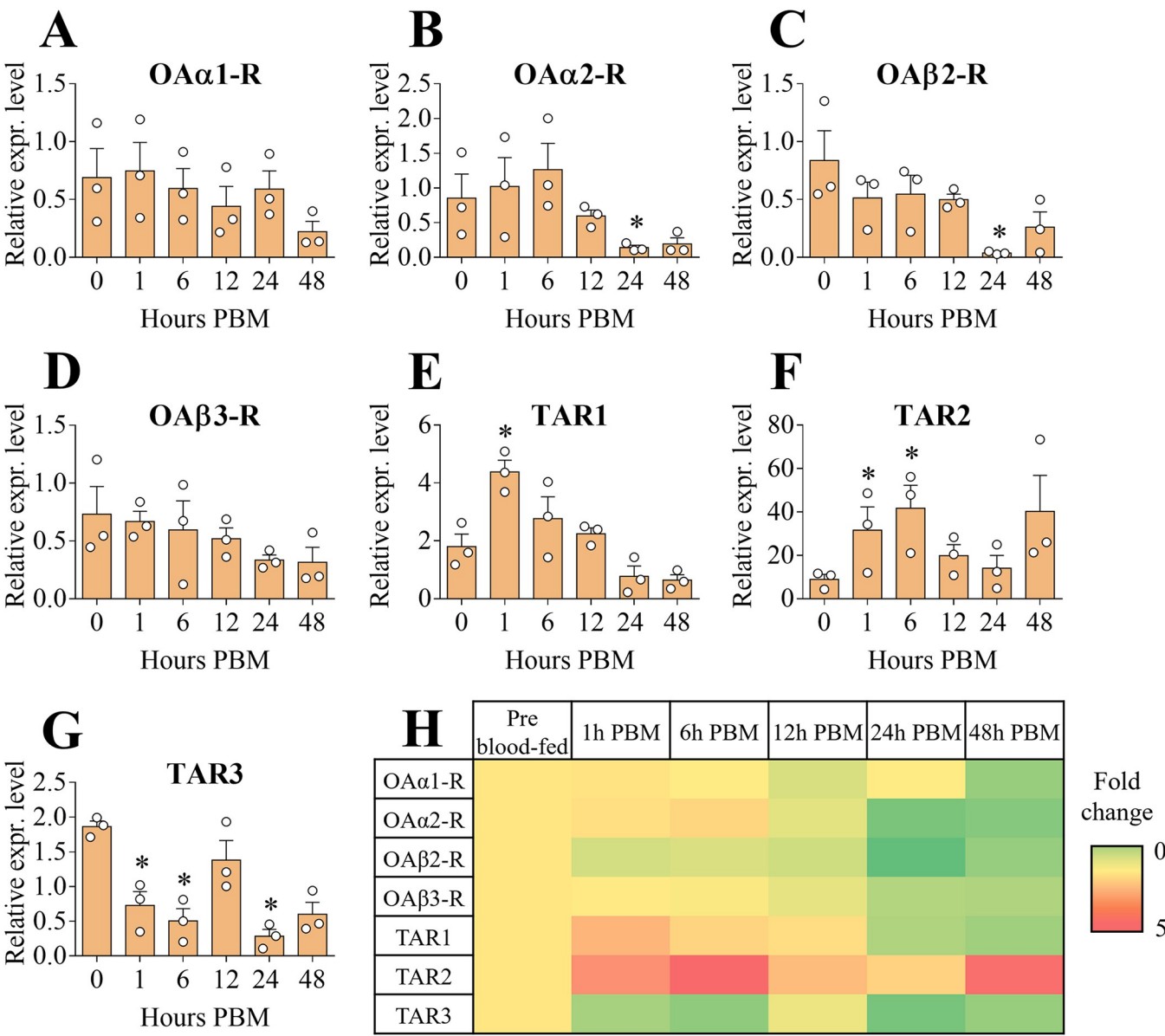

**Fig 7. Expression patterns of OA and TA receptor transcripts among female adult *A. aegypti* ovaries post blood meal (PBM) time points.** *AaOAα1-R* (**A**), *AaOAα2-R* (**B**), *AaOAβ2-R* (**C**), *AaOAβ3-R* (**D**), *AaTAR1* (**E**), *AaTAR2* (**F**) and *AaTAR3* (**G**). Setting all the receptor expression levels in pre blood-fed condition to 1, a heat map was assembled (**H**). Data represent means ± SEM of three biological replicates. * p<0.05 vs pre blood-fed condition (0 hours PBM) according to one-way ANOVA followed by multiple comparisons Bonferroni post hoc test.

abundance did not change across all the time point analysed (Fig 7D and 7H). With regards to TA receptors, *AaTAR1* showed a significantly increased abundance 1 hour PBM and a trend towards a reduction in abundance at 24 and 48 hours PBM (Fig 7E and 7H). *AaTAR2*, the receptor most abundantly expressed in ovaries (~10-fold) relative to all other TA or OA receptors in the pre blood-fed condition, showed a significantly increased abundance at 1 and 6 hours PBM. Furthermore, while not reaching a level of statistical significance, it was the only receptor with an increase in transcript abundance 48 hours PBM, suggesting a role in the control of *A. aegypti* reproduction (Fig 7F and 7H). On the hand, *AaTAR3* exhibited a significant

reduction in the mRNA abundance at 1, 6 and 24 hours PBM with similarly reduced levels at 48 hours PBM (Fig 7G and 7H).

### OA and TA synthesis in *A. aegypti*: Transcript expression pattern in developmental stages, adult tissues, and brain after a blood meal

The physiological and behavioural traits controlled by OA and TA receptors strongly depend on their activation by their aminergic ligands, OA and TA. For this reason, we investigated the expression profile of the gene transcripts encoding the enzymes responsible for TA and OA biosynthesis: tyrosine decarboxylase (*Tdc*) and tyramine β-hydroxylase (*Tβh*). In *A. aegypti* development stages, *Tdc* transcript was expressed most abundantly in eggs, with a significantly reduced mRNA abundance in 2$^{nd}$, 3$^{rd}$ and 4$^{th}$ instar larval stages. The *Tdc* expression levels then increased in pupae and in both male and female adults (Fig 8A). Conversely, *Tβh* transcript was significantly enriched only in adult males, suggesting a more prominent or sex-specific role of OA in adult male *A. aegypti* (Fig 8B). Both *Tdc* and *Tβh* transcripts were significantly enriched in adult male brain (Fig 8C and 8D). Interestingly, *Tdc* was more highly expressed than *Tβh* in MTs of adult mosquitoes (S8 Fig). To better understand how TA and OA biosynthesis might be affected by blood feeding, we investigated the temporal expression of both *Tdc* and *Tβh* transcripts in adult female brains, following a blood meal. *Tdc* trended towards an increased abundance at 1 and 6 hours PBM although this was not significant, followed by a reduction in the transcript abundance between 12 and 48 hours PBM (Fig 8E).

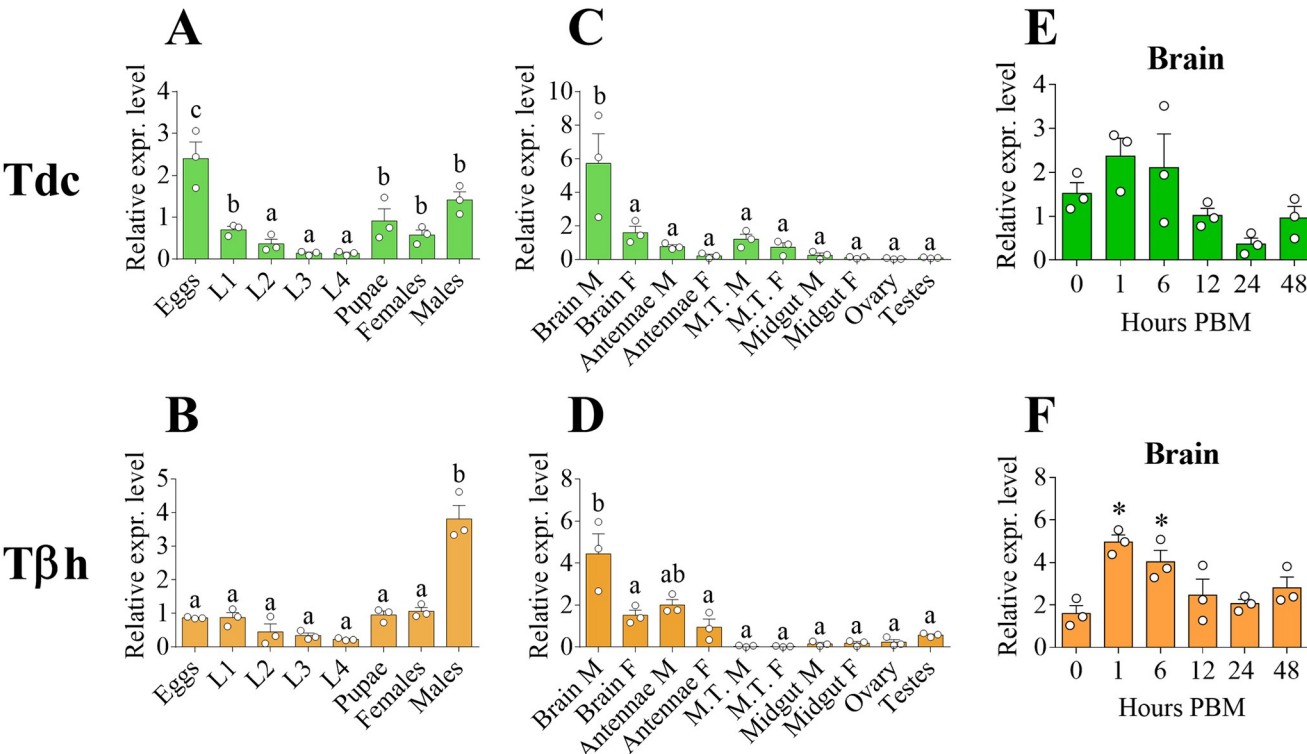

**Fig 8. Expression patterns of the enzymes involved in TA and OA biosynthesis: Tyrosine decarboxylase (*Tdc*) and tyramine β-hydroxylase (*Tβh*).** The transcript levels were investigated among development stages (**A, B**), several tissues collected from male and female adult mosquitoes (**C, D**) and female adult *A. aegypti* brains post blood meal (PBM) time points (**E, F**). Data represent means ± SEM of three biological replicates. Statistical significance of $p < 0.05$ is indicated by different letters in panels A-D. * $p < 0.05$ vs pre blood-fed condition (0 hours PBM) in panels E and F according to one-way ANOVA followed by multiple comparisons Bonferroni post hoc test. Malpighian tubules (MTs); male (M); female (F).

Comparatively, *Tβh* transcript exhibited a significant increase in transcript abundance at 1 and 6 hours PBM, followed by levels comparable to the pre blood-fed condition thereafter (Fig 8F).

## Discussion

Mosquitoes are responsible for the highest number of human deaths compared to all other animals, therefore identifying new control strategies of these disease vectors is a priority [7]. Understanding the physiology of mosquitoes may help identify new molecular and cellular targets for their management. Although *A. aegypti* have been studied for a long time, little is known about one of the most important signalling pathways present in invertebrates, namely the TA and OA systems. Since TA and OA receptors have emerged as interesting targets for both synthetic and natural insecticides [24, 31, 49], we have examined these receptors in the Dengue vector, *A. aegypti*.

We identified seven putative OA and TA receptor sequences in the *A. aegypti* genome. Bioinformatic analyses revealed two OA α-adrenergic like receptors (AaOAα1-R and AaOAα2-R), two OA β-adrenergic like receptors (AaOAβ2-R and AaOAβ3-R) and three TA receptors (AaTAR1, AaTAR2 and AaTAR3). In *D. melanogaster*, three OA β-adrenergic like receptors have been characterized: DmOAβ1-R, DmOAβ2-R and DmOAβ3-R [50]. However, although the Oaβ1-R sequences of several insects was used as query, we did not identify any OAβ1-R orthologs in the *A. aegypti* genome. Similarly, a neurotranscriptome analysis of *A. aegypti* [51] failed to identify an OAβ1-R, supporting the notion that *A. aegypti* mosquitoes lack this receptor class. In support of this view, when the OA receptors were characterized in *A. gambiae*, only two OA β-adrenergic like receptors (OAβ2-R and OAβ3-R) were identified [36]. It is therefore possible that the *Culicidae* more generally have only two OA β-adrenergic like receptors as they appear to lack OAβ1-R.

All seven *A. aegypti* OA and TA receptors share the characteristic seven transmembrane domains typical of the GPCR superfamily [52], along with phosphorylation and glycosylation sites that are essential for correct protein folding and receptor signalling [46, 53]. Furthermore, specific amino acid residues likely involved in the binding of OA and TA with their respective receptors were identified. For each receptor, an aspartic acid in the TM III and two to three serine residues in the TM V are associated with the receptors ability to bind TA and OA [54, 55]. However, a mutagenesis study performed on *A. gambiae* OA receptors showed that different residues localized in TM VI might also be involved in binding with TA and OA, revealing a possibly more complex ligand/receptor-coupling mechanism [36].

Although a developmental transcriptome of *A. aegypti* has been constructed [56], no information about the role of OA and TA receptors in mosquito growth and development were presented. We have found that all seven receptors are expressed throughout *A. aegypti* developmental stages. Except for *AaOAα1-R*, all other receptors showed an expression pattern characterized by high levels of transcript in eggs, followed by a decline in larval stages and then high levels in pupae and adults. A similar dynamic expression pattern was observed in *Plutella xylostella*, in which both OA and TA receptors showed the highest expression levels in eggs and pupae, compared with larvae and adult [57]. Furthermore, in *D. melanogaster*, *Nivaparlata lugens*, *Mythimna separata* and *A. gambiae*, the OA β-adrenergic like receptor expression profile over the developmental stages appears similar to that observed in *A. aegypti* [36, 49, 58, 59]. However, in *Chilo suppressalis*, *OAβ2-R* showed the highest level of expression in larvae compared to eggs, pupae, and adults, suggesting distinct expression profiles among the various insect species [30]. Although both OAα1-R and OAβ2-R are involved in *D. melanogaster* egg-laying by modulating oviduct muscle contraction, little is known about the role that OA and

its receptors play in embryonic development of the eggs [60–62]. Recently, it has been observed in *D. melanogaster* that OA, acting through the OAβ2-R receptor, regulates oocyte maturation and egg development inside the follicle, where it might be involved in oocyte quiescence [63].

Regarding TA receptors, *TAR1* has been found to be enriched in the egg in *Halyomorpha halys*, *D. melanogaster* and *P. xylostella* relative to other developmental stages [64–66]. Information about the role of TA and TAR1 in embryogenesis is limited to *D. melanogaster*, in which the *TAR1* gene transcript exhibits a dynamic expression pattern during embryo maturation inside the eggs, with a peak corresponding to nervous tissue formation [67]. In that study, the authors hypothesized that the decline in *TAR1* expression in larval stages may be explained by the relative decrease in the ratio of neuronal versus non-neuronal tissue. The high levels of expression of OA and TA receptors in *A. aegypti* eggs might still be necessary for egg maturation. Fuchs and colleagues [35] showed that injection of TA caused premature melanization of the eggs and a resulting egg retention in *A. gambiae*, supporting the notion that TA and OA signalling is critical for egg formation and maturation.

Since both OA and TA act as neurotransmitters in invertebrates, it is not surprising that the highest expression levels of all seven receptors are observed in the brains of adult *A. aegypti* compared to other examined tissues (reproductive organs, MTs, midgut, and antennae and rostrum). In adult male antennae and rostrum, *AaOAβ2-R* is the main receptor expressed, confirming what has been observed in the *A. aegypti* neurotranscriptome [51]. In mosquitoes, OA acts as a neurotransmitter and neuromodulator in antennae, involved in several physiological processes including sound detection [68] and olfaction [69]. Therefore, we suggest that OA might exert some of these functions in *A. aegypti* antennae by binding to AaOAβ2-R and influencing sensory perception. However, complex processes, such as pheromone perception, might be controlled by multiple OA and TA receptors working synergistically. OAα1-R [60] and TAR1 [24] also appear to be involved in olfactory perception in other insects [20].

The high expression of *AaTAR3* in both MTs and midgut raises the possibility for the involvement of TA signalling in *A. aegypti* osmoregulation and diuresis. The presence of a TA-mediated pathway was identified in the *D. melanogaster* stellate cells, one of the two main cell types comprising the MTs [70]. Investigating the expression profile of Tdc and Tβh, which encode for enzymes responsible for TA and OA biosynthesis, respectively, revealed that Tdc was more highly expressed than Tβh in MTs of adult mosquitoes (S8 Fig), suggesting that TA could indeed be synthesized in MTs and act as a modulator of diuresis. Together with the high *AaTAR3* transcript abundance observed in the MTs, this suggests the presence of a TA-mediated diuretic pathway in *A. aegypti* similar to *D. melanogaster*. Interestingly, however, in *D. melanogaster* the intracellular pathway for TA seems to involve mainly TAR2. Nevertheless, El-Kholy et.al. [65] observed that both DmTAR2 and DmTAR3 were expressed at high levels in *D. melanogaster* MTs and gut. Furthermore, it was later demonstrated that after DmTAR2 knock-down in MTs, a response to TA is still detectable, whereas knock-down of both DmTAR2 and DmTAR3 made MTs insensitive to TA [71]. Interestingly, the expression profile of TAR2 was investigated in different organs of adult *B. mori*, and no appreciable expression was observed in the MTs [72]. TA receptors are still poorly characterized and further studies are needed to better understand how TA signalling modulates the activity of MTs in insects.

As found here, high *AaTAR2* abundance in ovaries was also observed by Matthews and colleagues [51], suggesting that this TA receptor might be implicated in mosquito reproduction. Indeed, this possibility is supported by the observed enrichment of *AaTAR2* in the ovaries following a blood meal. However, TAR2 has not been well characterized in other insects and has not been highly investigated in reproductive organs.

A blood meal induces significant changes in the biology of the female mosquito. The blood is rich in nutrients that mosquitoes must absorb to ensure successful egg production [73]. In addition, it is essential for the mosquito to quickly excrete toxic elements, such as heme groups (reactive species that can induce oxidative stress) and maintain salt and water homeostasis [74, 75]. The midgut, MTs, and ovaries are major tissues involved in the above processes following a blood meal [76]. In general, after a blood meal, we observed a complex set of changes, characterized by dynamic profiles for each receptor in the brain, midgut, MTs, and ovaries. Surprisingly, blood feeding did not affect the expression profile of TA and OA receptors in the brain of female mosquitoes. On the other hand, transcripts encoding the enzymes responsible for the synthesis of TA and OA, respectively Tdc and Tβh, showed dynamic profiles, with peaks in the first hours after blood feeding, suggesting that blood intake induces a strong synthesis of both TA and OA. Furthermore, an increased synthesis of both TA and OA suggests that the subsequent physiological processes following blood feeding are influenced by these two biogenic amines that might act as neuromodulators and/or neurohormones. However, more experiments are needed to define the role of both TA and OA in regulating specific physiological traits in mosquitoes PBM.

In the ovaries, *AaTAR2* was the receptor that exhibited the highest expression relative to other TA and OA receptors and its abundance was further elevated at several time points analyzed after a blood meal, suggesting a crucial role in reproduction of *A. aegypti*. The role of TA and OA in insect reproduction has been demonstrated in several studies [23, 35, 77–80]. In two particular studies on *Apis mellifera* and *Locusta migratoria*, TAR1 was implicated as modulating the synthesis of vitellogenin by the fat body, the main protein source for egg growth [81, 82]. Perhaps a TA-mediated pathway involved in reproduction might be present in *A. aegypti*, via at least AaTAR2.

In both the MTs and midgut a complex scenario was observed. After a blood meal, most of the receptors for TA and OA showed changes in transcript abundance including both increased and decreased levels. In general, both AaTAR2 and AaTAR3 appear to be major receptor targets involved in controlling these organs, possibly influencing diuresis, osmoregulation, and blood digestion, although OA receptors might also participate in some of these processes.

## Conclusions

Overall, the identification and molecular characterization of four OA and three TA receptors in *A. aegypti* and their putative physiological roles based on transcript expression profile analysis will be useful for better understanding complex mechanisms such as reproduction and osmoregulation in this important vector of several human diseases. Furthermore, these receptors could be considered new targets for *A. aegypti* control. In order to develop innovative pest management strategies targeting TA and OA receptors, further experiments are needed, such as pharmacological profiles of the receptors, paying particular attention to their specific binding properties with various biogenic amines and other molecules.

## Supporting information

**S1 Fig. *Aedes aegypti* α1 octopamine receptor (AaOAα1-R).**
(DOCX)

**S2 Fig. *Aedes aegypti* α2 octopamine receptor (AaOAα2-R).**
(DOCX)

**S3 Fig.** *Aedes aegypti* **β2 octopamine receptor (AaOAβ2-R).**
(DOCX)

**S4 Fig.** *Aedes aegypti* **β3 octopamine receptor (AaOaβ3-R).**
(DOCX)

**S5 Fig.** *Aedes aegypti* **type 1 tyramine receptor (AaTAR1).**
(DOCX)

**S6 Fig.** *Aedes aegypti* **type 2 tyramine receptor (AaTAR2).**
(DOCX)

**S7 Fig.** *Aedes aegypti* **type 3 tyramine receptor (AaTAR3).**
(DOCX)

**S8 Fig. Tdc and Tβh expression profile in adult male and female mosquitoes Malpighian tubules.**
(DOCX)

**S1 Table. Primers used in this research work.**
(DOCX)

**S2 Table. Accession number of OA and TA receptors from different insect species used for the maximum likelihood phylogenetic analysis.**
(DOCX)

## Acknowledgments

The authors would like to thank Dr. Federica Albanese (Toronto Western Research Institute) for language revision and Dr. Jimena Leyria (University of Toronto Mississauga) for ongoing advice during the project.

## Author Contributions

**Conceptualization:** Luca Finetti, Jean-Paul Paluzzi, Ian Orchard, Angela B. Lange.

**Data curation:** Luca Finetti.

**Formal analysis:** Luca Finetti.

**Funding acquisition:** Jean-Paul Paluzzi, Ian Orchard, Angela B. Lange.

**Investigation:** Luca Finetti.

**Methodology:** Luca Finetti, Jean-Paul Paluzzi, Ian Orchard.

**Project administration:** Ian Orchard, Angela B. Lange.

**Resources:** Jean-Paul Paluzzi, Ian Orchard, Angela B. Lange.

**Supervision:** Jean-Paul Paluzzi, Ian Orchard, Angela B. Lange.

**Validation:** Luca Finetti.

**Writing – original draft:** Luca Finetti.

**Writing – review & editing:** Luca Finetti, Jean-Paul Paluzzi, Ian Orchard, Angela B. Lange.

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
