## [Decision Letter · Decision Letter 0]

24 Jan 2023

PONE-D-22-34361Octopamine and tyramine signalling in Aedes aegypti: molecular characterization and insight into potential physiological roles.PLOS ONE

Dear Dr. Finetti,

Thank you for submitting your manuscript to PLOS ONE. After careful consideration, we feel that it has merit but does not fully meet PLOS ONE’s publication criteria as it currently stands. Therefore, we invite you to submit a revised version of the manuscript that addresses the points raised during the review process. In particular, methodological questions were raised regarding the statistical analyses and greater clarity was requested regarding the expression profile interpretation of the biosynthetic enzymes. 

We look forward to receiving your revised manuscript.

Kind regards,

J Joe Hull, Ph.D.

Academic Editor

PLOS ONE

Journal Requirements:

"The authors would like to thank Dr. Federica Albanese (Toronto Western Research Institute) for language revision and Dr. Jimena Leyria (University of Toronto Mississauga) for ongoing advice during the project. This research was supported by NSERC Discovery Grants to JPP, IO and ABL. "

"This research was supported by NSERC Discovery Grants to JPP, IO and ABL. "

Reviewers' comments:

Reviewer's Responses to Questions

**Comments to the Author**

1. Is the manuscript technically sound, and do the data support the conclusions?

Reviewer #1: Yes

Reviewer #2: Yes

2. Has the statistical analysis been performed appropriately and rigorously? 

Reviewer #1: Yes

Reviewer #2: Yes

3. Have the authors made all data underlying the findings in their manuscript fully available?

Reviewer #1: Yes

Reviewer #2: Yes

4. Is the manuscript presented in an intelligible fashion and written in standard English?

Reviewer #1: Yes

Reviewer #2: Yes

5. Review Comments to the Author

Reviewer #1: The manuscript gives a good contribution in the identification and molecular characterization of four OA and three TA receptors in A. aegypti and their putative physiological roles based on transcript expression profile analysis will be useful for better understanding complex mechanisms such as reproduction and osmoregulation in this important vector of several human diseases. However, I have concerns about the methods, results and discusssion of the ms that I believe need to be addressed in order to improve its clarity. Their approach is interesting but it has some flaws that make this version unacceptable for publication.

My main concern:

1- Line 122: Why did you choose 12/12 instead of 8/16 D/L cycle?

2- Line 301: I would suggest to put subheading as a sentence instead of question form.

3- Line 303: What do authors mean with (Female A. aegypti require a blood meal for egg production and it is also true that this process imposes some stress on the insect?)

4- Line 316: Unfed means starved or sugar-fed? First, because it is well known that starvation stimulate stess hormone, octopamine, the counterpart of adrenaline in vertebrates and consequently alter the expression level of the corresponding receptors. Second, did authors compare their control at the same time points?

5- Heat maps included in figure (4-7) did not add more information especially no significant difference among chosen time points are found.

6- Line 422: The expression level of Tβh in males is highly upregulated, however, tests (the only exclusive organ to males among all chosen organs) show very low expression level, from where this difference appear? How?

7- Lines 484-486: According to Fig.8, this upregulation was not statistically significant, you should not magnify your results if they are not statistically significant. I suggest to focus on the most important and statistically significant data in the whole ms.

8- Line 501: Did you compare statistically the expression level of tdc with the expression level of Tβh? If not, how did you conclude that tdc was more highly expressed than Tβh in MTs?

9- Line 517-518: Regarding the sentence (and has not been highly investigated in reproductive organs but was shown to be present in D. melanogaster female reproductive organs (El-Kholy et al., 2015).

I suggest to delete this sentence because El-Kholy et al (2015) detect very weak TyRII expression level in male and female reproductive system (almost nothing) but the strongest expression of this receptor was detected in intestine and MTs.

10- Lines (526-530): Regarding the sentence (Surprisingly, blood feeding did not affect the expression profile of TA and OA receptors in the brain of female mosquitoes. On the other hand, transcripts encoding the enzymes responsible for the synthesis of TA and OA, respectively Tdc and Tβh, showed dynamic profiles, with peaks in the first hours after blood feeding, suggesting that blood intake induces a strong synthesis of both TA and OA), This means that in female mosquitoes, blood meal induce more OA and TA to be synthetized but without any consequent change in their related receptors, please explain.

11- Line 505: Revise the written mistakes in all references in the text for example replace (El-Kholi et al. 2015) with (El-Kholy et al. 2015).

Reviewer #2: The authors have included most of my recommendations. However, there are a few concerns that have not been addressed:

Introduction:

Line 68: I would not consider Wolbachia a genetic technique. Perhaps, authors meant “genetic techniques or Wolbachia infected mosquitoes”?

Results:

Line 214: Introduce abbreviation “Oaα2”

In the figures, it is still not clear to me which significance levels a-b mean, and which group they are compared to, does the statistics refer to comparisons to the reference genes?. Please, explain. Did they use Bonferroni, Dunnet or Tukey methods? The explanation is not consistent in the Methods and figure captures.

Results: “…associated with smell and taste” and hearing? I do not think the references has been included:

Georgiades M, Alampounti CA, Somers J, Su M, Ellis D, Bagi J, et al. A novel beta-adrenergic like octopamine receptor modulates the audition of malaria mosquitoes and serves as insecticide target [Internet]. bioRxiv; 2022 [cited 2022 Sep 13]. p. 2022.08.02.502538.

This reference can also be discussed in 491-495.

6. PLOS authors have the option to publish the peer review history of their article (what does this mean?). If published, this will include your full peer review and any attached files.

Reviewer #1: **Yes: **Samar El-Kholy

Reviewer #2: No

---

## [Author Response · Author response to Decision Letter 0]

1 Feb 2023

Response to reviewers: manuscript PONE-D-22-34361

Reviewer 1:

The manuscript gives a good contribution in the identification and molecular characterization of four OA and three TA receptors in A. aegypti, and their putative physiological roles based on transcript expression profile analysis will be useful for better understanding complex mechanisms such as reproduction and osmoregulation in this important vector of several human diseases. However, I have concerns about the methods, results, and discussion of the MS that I believe need to be addressed in order to improve its clarity. Their approach is interesting, but it has some flaws that make this version unacceptable for publication.

My main concern:

1- Line 122: Why did you choose 12/12 instead of 8/16 D/L cycle?

Reply: For Aedes aegypti rearing we followed the “Guidelines for routine colony maintenance of Aedes mosquito species” by FAO (Food and Agriculture Organization of the United Nations) that suggests 12:12 as D/L cycle. Furthermore, we followed the rearing conditions of the colony from which we obtained the animals as described in previous studies (Wahedi & Paluzzi, 2018).

Now, the text reads (lines 122-123): All mosquitoes were raised under a 12:12 light: dark cycle at 26 °C as previously described (Wahedi & Paluzzi, 2018).

Reference: 

Wahedi, A., Paluzzi, J.P (2018). Molecular identification, transcript expression, and functional deorphanization of the adipokinetic hormone/corazonin-related peptide receptor in the disease vector, Aedes aegypti. Sci Rep 8, 2146.

2- Line 301: I would suggest putting subheading as a sentence instead of question form.

Reply: The subheading has been changed. Now the subheading reads (line 301): “TA and OA receptor transcript expression patterns after a blood meal.”

3- Line 303: What do authors mean with (Female A. aegypti require a blood meal for egg production and it is also true that this process imposes some stress on the insect?)

Reply: The aim of this sentence was highlighting the changes that occur in mosquito physiology and behavior after a blood meal. Although the blood is required for egg production, mosquitoes need to eliminate toxins present in the blood and rapidly balance the water and salts amount. Together, these processes impose a stress onto the insect.

4- Line 316: Unfed means starved or sugar-fed? First, because it is well known that starvation stimulate stress hormone, octopamine, the counterpart of adrenaline in vertebrates and consequently alter the expression level of the corresponding receptors. Second, did authors compare their control at the same time points?

Reply: We thank the reviewer for this comment. In these cases, unfed had referred to sugar-fed (but non-blood fed). No, we did not compare the data with their control at the same time points. We clarify the question in the manuscript. Now the text reads (lines 123-124): “All adult mosquitoes were fed with 10% sucrose (w/v) ad libitum and used for experiments three days-post ecdysis”. Furthermore, to avoid any confusion, the term “unfed” has been replaced with “pre blood-fed” across the manuscript. 

5- Heat maps included in figure (4-7) did not add more information especially no significant difference among chosen time points is found.

Reply: Thanks to the reviewer for the advice. However, we believe that the heatmaps will help the readers to better visualize the data in a general view and better identify the alterations in TA and OA receptor expression levels. In setting all the receptor expression levels to pre blood-fed condition to 1 allows us to observe which receptors are more affected in terms of transcript levels after a blood meal. 

6- Line 422: The expression level of Tβh in males is highly upregulated, however, tests (the only exclusive organ to males among all chosen organs) show very low expression level, from where this difference appear? How?

Reply: We believe that the difference in expression profile between male and female mosquitoes might be related to the brain, tissue in which male mosquitoes exhibit a higher amount of transcript compared to the females. The point has been previously mentioned in the manuscript (lines 410-411). 

7- Lines 484-486: According to Fig.8, this upregulation was not statistically significant, you should not magnify your results if they are not statistically significant. I suggest focusing on the most important and statistically significant data in the whole ms.

Reply: This sentence refers to the expression pattern of the TA and OA receptors represented in Figure 3, not the enzymes Tdc and Tβh shown in Figure 8. 

8- Line 501: Did you compare statistically the expression level of tdc with the expression level of Tβh? If not, how did you conclude that tdc was more highly expressed than Tβh in MTs?

Reply: We compared the expression profile of tdc and Tβh in MTs (the graph is now available as Supplementary Figure S8) and the expression levels of Tdc in both male and female MTs are higher compared to the Tβh levels. We added some clarification in the revised manuscript:

Lines 410-411: “Interestingly, Tdc was more highly expressed than Tβh in MTs of adult mosquitoes (Supplementary Figure S8).”

Lines 502-505: “Investigating the expression profile of Tdc and Tβh, which encode for enzymes responsible for TA and OA biosynthesis, respectively, revealed that Tdc was more highly expressed than Tβh in MTs of adult mosquitoes (Supplementary Figure S8), suggesting that TA could indeed be synthesized in MTs and act as a modulator of diuresis.”

Supplementary Figure S8:

Supplementary Figure S8: Transcript expression of genes encoding the enzymes involved in TA and OA biosynthesis: tyrosine decarboxylase (Tdc) and tyramine β-hydroxylase (Tβh). The transcript levels were investigated in Malpighian tubules of adult male and female mosquitoes. Data represent means ± SEM of three biological replicates. Statistical significance is indicated by ** p<0.01 according to one-way ANOVA followed by multiple comparisons Bonferroni post hoc test. Malpighian tubules (MTs).

9- Line 517-518: Regarding the sentence (and has not been highly investigated in reproductive organs but was shown to be present in D. melanogaster female reproductive organs (El-Kholy et al., 2015).

I suggest deleting this sentence because El-Kholy et al (2015) detect very weak TyRII expression level in male and female reproductive system (almost nothing) but the strongest expression of this receptor was detected in intestine and MTs.

Reply: Thank you. We have fixed the sentence. Now the text reads (lines 5208-521): “However, TAR2 has not been well characterized in other insects”.

10- Lines (526-530): Regarding the sentence (Surprisingly, blood feeding did not affect the expression profile of TA and OA receptors in the brain of female mosquitoes. On the other hand, transcripts encoding the enzymes responsible for the synthesis of TA and OA, respectively Tdc and Tβh, showed dynamic profiles, with peaks in the first hours after blood feeding, suggesting that blood intake induces a strong synthesis of both TA and OA), This means that in female mosquitoes, blood meal induce more OA and TA to be synthetized but without any consequent change in their related receptors, please explain.

Reply: The variation in TA and OA synthesis does not necessarily reflect changes in TA and OA receptor expression in the brain. In this case, the blood meal induces changes in TA and OA production which in turn may act as neuromodulators and/or neurohormones in tissues distant from the CNS.

We clarify more in detail this aspect in the discussion. Now the text reads (lines 529-536): “Surprisingly, blood feeding did not affect the expression profile of TA and OA receptors in the brain of female mosquitoes. On the other hand, transcripts encoding the enzymes responsible for the synthesis of TA and OA, respectively Tdc and Tβh, showed dynamic profiles, with peaks in the first hours after blood feeding, suggesting that blood intake induces a strong synthesis of both TA and OA. Furthermore, an increased synthesis of both TA and OA suggests that the subsequent physiological processes following blood feeding are influenced by these two biogenic amines that might act as neuromodulators and/or neurohormones.”

11- Line 505: Revise the written mistakes in all references in the text for example replace (El-Kholi et al. 2015) with (El-Kholy et al. 2015).

Reply: Thank you. All the references across the text have been checked and fixed.

Reviewer 2:

Introduction:

Line 68: I would not consider Wolbachia a genetic technique. Perhaps, authors meant “genetic techniques or Wolbachia infected mosquitoes”?

Reply: Thank you, we fixed the sentence. Now the text reads (lines 67-69): “The principal intervention strategies include the use of insecticides (such as pyrethroids), genetic techniques (sterile male) or using Wolbachia infected mosquitoes (McGregor & Connelly, 2021).”

Results:

Line 214: Introduce abbreviation “Oaα2”

Reply: Thank you. Oaα2 is a typographical error. It referred to OAα2 receptors. We have fixed all the errors across the manuscript (e.g., line 214).

In the figures, it is still not clear to me which significance levels a-b mean, and which group they are compared to, does the statistics refer to comparisons to the reference genes? Please, explain. Did they use Bonferroni, Dunnet or Tukey methods? The explanation is not consistent in the Methods and figure captures.

Reply: We have now clarified the statistical analysis across all the figure legends and material & methods. Now the text reads (lines 183-185): “Data are expressed as means ± s.e.m. of n experiments and were analyzed using one-way analysis of variance (ANOVA) followed by Bonferroni test for multiple comparison.”

Results: “…associated with smell and taste” and hearing? I do not think the references has been included:

Georgiades M, Alampounti CA, Somers J, Su M, Ellis D, Bagi J, et al. A novel beta-adrenergic like octopamine receptor modulates the audition of malaria mosquitoes and serves as insecticide target [Internet]. bioRxiv; 2022 [cited 2022 Sep 13]. p. 2022.08.02.502538.

This reference can also be discussed in 491-495.

Reply: Thank you to the reviewer for the suggestion but we would avoid referencing preprints.

---

## [Editor Report · Decision Letter 1]

5 Feb 2023

Octopamine and tyramine signalling in Aedes aegypti: molecular characterization and insight into potential physiological roles.

PONE-D-22-34361R1

Dear Dr. Finetti,

We’re pleased to inform you that your manuscript has been judged scientifically suitable for publication and will be formally accepted for publication once it meets all outstanding technical requirements.

Kind regards,

J Joe Hull, Ph.D.

Academic Editor

PLOS ONE
---

## [Editor Report · Acceptance letter]

8 Feb 2023

PONE-D-22-34361R1 

Octopamine and tyramine signalling in *Aedes aegypti*: molecular characterization and insight into potential physiological roles. 

Dear Dr. Finetti:

I'm pleased to inform you that your manuscript has been deemed suitable for publication in PLOS ONE. Congratulations! Your manuscript is now with our production department. 

Kind regards, 

on behalf of

Dr. J Joe Hull 

Academic Editor

PLOS ONE